# Implicit Semantic Response Alignment for Partial Domain Adaptation

**Wenxiao Xiao**
Department of Computer Science
Brandeis University
Waltham, MA 02451
`wenxiaoxiao@brandeis.edu`

**Zhengming Ding**
Department of Computer Science
Tulane University
New Orleans, LA 70118
`zding1@tulane.edu`

**Hongfu Liu**
Department of Computer Science
Brandeis University
Waltham, MA 02451
`hongfuliu@brandeis.edu`

## Abstract

Partial Domain Adaptation (PDA) addresses the unsupervised domain adaptation problem where the target label space is a subset of the source label space. Most state-of-art PDA methods tackle the inconsistent label space by assigning weights to classes or individual samples, in an attempt to discard the source data that belongs to the irrelevant classes. However, we believe samples from those extra categories would still contain valuable information to promote positive transfer. In this paper, we propose the Implicit Semantic Response Alignment to explore the intrinsic relationships among different categories by applying a weighted schema on the feature level. Specifically, we design a class2vec module to extract the implicit semantic topics from the visual features. With an attention layer, we calculate the semantic response according to each implicit semantic topic. Then semantic responses of source and target data are aligned to retain the relevant information contained in multiple categories by weighting the features, instead of samples. Experiments on several cross-domain benchmark datasets demonstrate the effectiveness of our method over the state-of-the-art PDA methods. Moreover, we elaborate in-depth analyses to further explore implicit semantic alignment.

## 1 Introduction

Deep neural networks have achieved impressive results in various supervised learning applications such as object recognition [15, 18], semantic segmentation [51, 12, 48] and multi-modal learning [34, 40], but annotating a large-scale dataset for deep learning models training could be tremendously grueling and expensive. Thus, considerable efforts have been dedicated to domain adaptation (DA), which attempts to circumvent labeling unfamiliar target data by transferring knowledge from a well-studied source domain data. Existing DA methods align two differently-distributed domains by finding invariant representations from the transferable feature structure[24]. Matching the source and target distributions with a discrepancy loss based on high-order statistics such as maximum mean discrepancy (MMD) [1] is commonly used along with iterative pseudo pseudo labeling strategy [16, 37, 22]. Recently efforts seek a domain-invariant feature space by adversarial learning, so that the source classifier be directly used for the target data prediction. Adding a weighting schema further improves the classification ability of adversarial methods. However, traditional domain adaptation

requires a related source domain that shares the same label space with the unlabelled target domain, which is not always available in real-life. Sometimes, target label space is only a subset of source labels space when people try to transfer knowledge from a more comprehensive domain, and partial domain adaption (PDA) is introduced because of this practical situation.

The mismatched label space poses a difficult challenge where aligning all source domain with the small target domain could suffer from negative transfer problem. To this end, various PDA algorithms [2, 50, 3, 14, 19, 4, 17, 37, 23] extend the reweighting schema to alleviate negative transfer by reducing the influence of the irrelevant categories. Down-weighting the samples from the outlier source classes is one of the most commonly used strategies, while other recently works further assign weights on the instance level for both source and target samples to mitigate domain divergence. In general, these PDA methods aim to identify and discard all the irrelevant classes, in an attempt of aligning the marginal distributions only for the categories shared both domains. However, some extra classes are semantically correlated with the target classes, which could potentially contain valuable information for the PDA task. We believe that fully utilizing the relevant information hidden across different classes helps promote positive transfer. For example, cats and dogs have clear distinguished features for class separation; however, they also share many common semantic topics including fur, four legs, and so on. Therefore, we expect to explore the relationships among different categories by implicit semantics and achieve the semantic level alignment.

**Contributions**. In this paper, we propose the implicit semantic response alignment, as a plug-in module for any existing PDA models, which extracts implicit semantics from all categories including the shared categories and source-only categories and reduce the distribution discrepancy on the semantic level. Specifically, every sample is similarly decomposed into an embedding vector representing diverse implicit semantic topics with a class2vec machine. Under the guidance of each implicit semantic topic, we employ an attention-based weighting schema on the features by the implicit semantic response. The semantically weighted feature masks of the source and target are calibrated together as our final alignment. The major contributions are summarized as follows:

- We exploit the relationships of all available categories by extracting implicit semantics from visual features, which allows our method to utilize relevant information contained in every sample, including those from the long neglected unshared categories in the PDA problem.
- With the help of a novel feature-level weighting strategy guided by implicit semantic responses, we align the source and target data distribution based on the implicit semantic topics shared between two domains to boost the positive transfer.
- We demonstrate the effectiveness of our method by boosting the existing state-of-art PDA models on various benchmarks including *Office31* [36], *Office-Home* [44] and *ImageNet-Caltech* [4], and provide several detailed in-depth explorations of our purposed method.

## 2 Related Work

Unsupervised domain adaptation (UDA) has attracted increasing attention over the past decades due to the high cost of annotating a massive amount of new data for training deep convolutional neural networks. Various UDA models have been established to overcome the distribution disparities between the source and target domains, among which two types of strategies are widely adopted. The first type [42, 28, 25, 8, 27, 30] matches two data distributions based on their high-order statistical properties to diminish domain discrepancy, such as maximum mean discrepancy (MMD) [1]. Inspired by Generative Adversarial Networks [9], other UDA methods [6, 7, 41] resort to adversarial learning, which aims to capture a domain-invariant feature representation that is capable of confusing the domain discriminator. Some works [33, 26, 39] also incorporate reweighting schema derived from the interaction between discriminator and classifier to further align different domains. Additionally, iterative pseudo labeling is used by recent methods [5, 22, 46] in an attempt to align the label spaces of different domains. However, in a more realistic situation where the source and target label spaces are not identical, the prediction capability of these methods is seriously impaired by negative transfer due to the heterogeneous label distributions.

A significant amount of research efforts [13, 31, 29, 21, 20, 49, 38] have been done on diverse aspects of transferring knowledge between domains with mismatched label space, while this work only focuses on a special case, partial domain adaptation (PDA) where the source label space subsumes target label space. Most conventional PDA methods [2, 50, 3, 14, 19, 4, 17, 37, 23] extend the deep

adversarial adaptation introduced by Domain-Adversarial Neural Network [7] with varied reweighting approaches. Selective Adversarial Network (SAN) [2] employs multiple class-wise discriminators to select out the source samples in the outlier classes and down-weight their importance during transfer. Similarly, Importance Weighted Adversarial Nets (IWAN) [50] assesses the importance of source samples by one auxiliary domain classifier along with a fixed source feature extractor. Partial Adversarial Domain Adaptation (PADA) [3] assigns class-level weights for both source classifier and domain adversary according to the estimated target label distribution. Multi-Weight Partial Domain Adaptation [14] exploits a multi-weight mechanism consisting of shared-class weights and shared-sample weights calculated by a shared-sample classifier to distinguish outlier classes and samples. Deep Residual Correction Network (DRCN) [19] utilizes a weighted class-wise matching strategy with a plug-in residual block that automatically identifies the most relevant source subclasses and aligns with them target data. Example Transfer Network (ETN) [4] weights source samples based on their discriminative information extracted by a transferability quantifier. Adaptively-Accumulated Knowledge Transfer ($A^2$TK) [17] optimizes both domain-wise distribution adaptation and class-wise distribution alignment by iteratively filtering out confident task-relevant target samples and their corresponding source categories. Select, Label, and Mix (SLM) [37] integrates pseudo labelling for the target domain to improve the discriminability of the invariant feature representation shared by the source and target domain. Lately, $BA^3US$ [23] incorporates weighting schema with augmenting target domain with source data, which achieves state-of-art on several PDA benchmark datasets. Generally, existing PDA methods involving adversarial training aim to transform the PDA task into an UDA-like problem by matching the source and target data on the sample level. On the other hand, our method tries to transform the problem on the semantic level, then align two different domains based on the implicit semantics. Aside from adversarial learning, recent UDA methods adopting other adaptation approaches also yield superior results for some PDA benchmarks. Larger Norm More Transferable (AFN) [47] accomplishes alignment by reducing a novel statistic distance designed by the authors that characterizes the mean-feature-norm discrepancy between different domains. Source Hypothesis Transfer (SHOT) [22] proposes a source data-free framework that learns target-specific features that can be accurately recognized by the frozen classifier trained on the source domain, with the help of self-supervised pseudo-labeling and information maximization.

Different from the existing literature, we propose the implicit semantic response alignment, a plug-in module to add on the existing PDA frameworks, to explore the relationships among different categories. Rather than the category level transfer, our semantic level alignment gathers all semantic response from multiple related categories across source and target domains to boost positive transfer.

# 3 Methodology

In this section, we elaborate on our method for partial domain adaptation. First, the overview of the proposed framework is introduced. Then we provide detailed technical formation and objective function for each component in our model.

## 3.1 Framework Overview

To achieve implicit semantic alignment, we propose a novel framework as shown in Figure 1 that consists of implicit semantic discovery and semantic alignment. The implicit semantic discovery module extracts diverse semantics from the backbone features with a class2vec machine, where every data point is represented by an embedding vector. Each dimension in this semantic space is considered as a implicit semantic topic, which guides the following source and target feature space alignment as an intermediate signal. Based on the discovered implicit semantics, we firstly calculate the response of every implicit topic to the feature space with an attention layer. Specifically, we obtain the attention signal of the backbone, such as ResNet, corresponding to every implicit semantic topic with a topic attention receptor, and assign attention weights to the features based on the signal strength. By taking the dot product of the attention weights and backbone features, we obtain a weighted feature mask to each implicit topic. The final semantic response alignment is achieved by forcing the source and target mask vectors corresponding to the same implicit topic to be as similar as possible. Implicit semantic guided features allow our model to transfer the relevant semantic information contained by the samples from the extra classes with similarity in certain semantics and boost the positive transfer. In the following section, we elaborate the implicit semantic response alignment details.

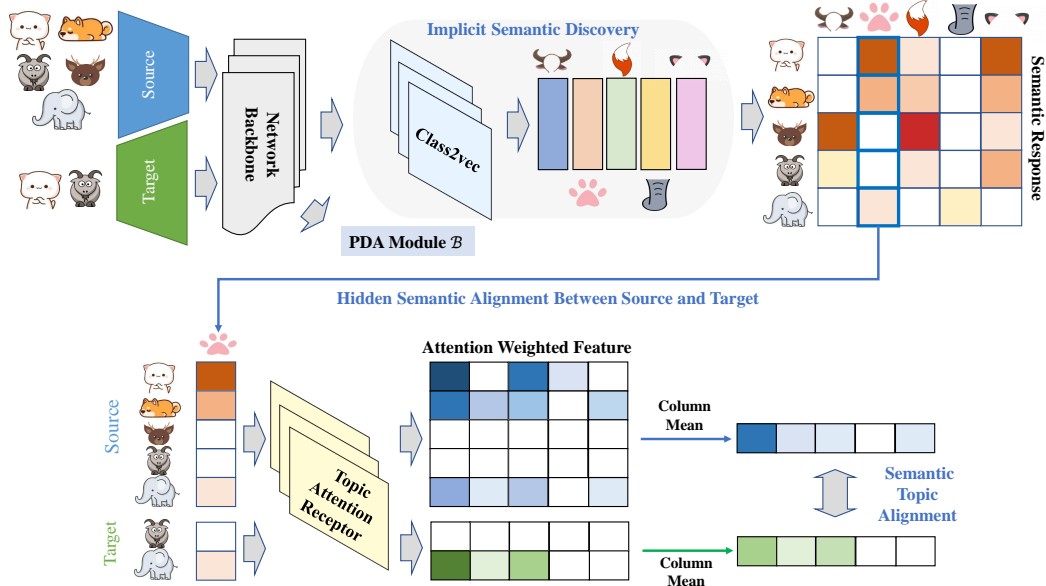

Figure 1: Framework of our proposed implicit semantic response alignment, where the upper parts demonstrate the class2vec layers discover implicit semantics and convert each sample to a semantic vector, while the lower parts illustrate the topic attention receptors calculate the weighted feature masks and the semantic alignment layers align source and target semantic response based on each implicit semantic topic for backbone feature extractor update.

## 3.2    Implicit Semantic Discovery and Alignment

Given a source domain with $n_s$ labeled samples associated with $\mathcal{C}_s$ categories and a target domain of $n_t$ unlabeled samples associated with $\mathcal{C}_t$ categories, Partial Domain Adaptation (PDA) occurs when the source label space subsumes the target label space, i.e., $\mathcal{C}_t \subset \mathcal{C}_s$. To tackle the partial domain adaptation, a PDA model $\mathcal{B}$ extracts visual features and reduces the feature dimension down to $d$ with a bottleneck layer. On this $d$-dimensional space, $\mathcal{B}$ then matches the labeled source domain $\mathcal{D}^s = \{X^s, Y^s\}$ and unlabelled target domain $\mathcal{D}^t = \{X^t\}$. To further improve exsiting PDA methods, we propose the class2vec machine to seek the implicit semantics shared by various categories and align the semantic response between source and target data. Let $f$ denote the class2vec machine and $g_j$ is the topic attention receptor for the $j$-th dimension of implicit semantic space $\mathcal{Z}$. To be exact, $f$ extracts the implicit semantic embedding from the feature space of a certain backbone PDA model $\mathcal{B}$ and $g_j$ calculates the response of $X$ to the $j$-th implicit semantic topic, i.e., $f : \mathcal{X} \to \mathcal{Z} \subseteq \mathbb{R}^{d_e}$, $g_j : \mathcal{X} \to \mathcal{Z}_j \subseteq \mathbb{R}$, for $j \in (1, ..., d_e)$, where $d_e$ is the number of implicit semantic topics. The semantic topic extractor, as well as the topic attention receptors, are shared by both source and target domains.

**Implicit semantic discovery**. We use an auto-encoder to achieve the implicit semantic topic extraction from the backbone feature space $X$. The auto-encoder consists of the class2vec machine, also known as encoder $f$ and decoder $h : \mathcal{Z} \to \mathcal{X} \subseteq \mathbb{R}^d$ which maps the hidden space back to a reconstruction of the original features. This implicit semantic discovery module is trained on the $l_2$ reconstruction error term as follows:

$$\mathcal{L}_{c2v} = \frac{1}{n_s} \sum_{i=1}^{n_s} ||x_i^s - h \circ f(x_i^s)||_2^2 + \frac{1}{n_t} \sum_{i=1}^{n_t} ||x_i^t - h \circ f(x_i^t)||_2^2, \tag{1}$$

where $\circ$ denotes the function composition.

**Topic attention reception**. In order to capture the interaction between the backbone features and each implicit semantic topic, we employ a topic attention receptor $g_j$ to regress $X$ on $Z_j$, $j \in (1, ..., d_e)$. The absolute gradients of $g_i$'s input layer are treated as the attention signals of the backbone features corresponding to the $j$-th semantic topic. We add $l_1$ regularization on the first layer to encourage

sparsity of the gradients. The total prediction loss for all $d_e$ regression models is expressed as:

$$\mathcal{L}_{reg} = \frac{1}{d_e} \sum_{j=1}^{d_e} \Big( \sum_{i=1}^{n_s} ||f(x_i^s)_j - g_j(x_i^s)||_2^2 + \sum_{i=1}^{n_t} ||f(x_i^t)_j - g_j(x_i^t)||_2^2 + \lambda_{reg} \sum_{p=1}^{P} ||w_{j,p}||_1 \Big), \quad (2)$$

where $f(x)_j$ donates the $j$-th dimension of vector $f(x)$, and $w_{j,p}$ donates the $p$-th weight in the first layer of $g_j$, which totally contains $P$ activations in its first layer. Note that we only need the input gradients, so these multilayer perceptron regressors are trained independently from other parts of the model. Thus $\mathcal{L}_{reg}$ does not contribute to the total alignment loss.

**Semantic topic alignment**. We encourage the semantics from source and target domains as similar as possible with semantic topic alignment. With the topic-specific attention, we further assign weights to the backbone features based on the semantic attention signals. For the $j$-th hidden topic, $A^{(j)} \in \mathbb{R}^{n \times d}$, which is the absolute gradients of the $i$-th topic attention receptor $g_i$ with respect to the input feature $X$, is used as the attention weights of $X$. The attention feature mask for the $j$-th semantic topic is defined as $M_j = N(A_j) \otimes X$, where $\otimes$ denotes element-wise multiplication, and $N(\cdot)$ denotes $l_2$ normalization operator. The source/target attention feature masks $M_j^{s/t}$ are the sub-matrices of $M_j$ corresponding to the source/target features $X_{s/t}$, respectively. The final semantic alignment is guided by the loss term as follows:

$$\mathcal{L}_a = \sum_{j=1}^{d_e} ||\text{sum}(M_j^s, 1) - \text{sum}(M_j^t, 1)||_2^2, \quad (3)$$

where $\text{sum}(\cdot, 1)$ returns a row vector containing the sum of each column.

**Overall Objective**. Based on a backbone PDA model $\mathcal{B}$, we propose our overall objective function as follows:

$$\min_{\mathcal{B},f,h} \mathcal{L}_{\mathcal{B}} + \alpha \mathcal{L}_{c2v} + \beta \mathcal{L}_a, \quad \min_{g_j} \mathcal{L}_{reg}, \quad (4)$$

where $\mathcal{L}_{\mathcal{B}}$ is the loss function of the backbone PDA model $\mathcal{B}$, and $j \in (1, ..., d_e)$. We use hyper-parameters $\alpha$ and $\beta$ to balance between semantic discovery and alignment, respectively. The topic attention receptors $g_j$ are trained independently and do not propagate to other parts of the network.

# 4 Experiments

In this section, we demonstrate the performance of our proposed implicit semantic response alignment by comparing with several state-of-the-art partial domain adaptation methods. We first introduce the experimental setup, report the algorithmic performance and finally provide several in-depth exploration of our method at various perspectives.

## 4.1 Experimental Setup

**Datasets**. We include three domain adaptation benchmark datasets for performance evaluation. (1) *Office-Home* dataset [44] is a challenging benchmark that includes four domains: Artistic images (Ar), Clip Art images (Cl), Product images (Pr) and Real World image (Rw). All four domains contain totally 65 object categories, and we likewise follow PADA and select the first 25 categories (in alphabetic order) within each domain as partial target domain. (2) *Office31* dataset [36] contains images of 31 object categories in three domains: Amazon (A), DSLR (D) and Webcam (W). These 31 categories are commonly encountered in office environment. Following the same data protocol as PADA [3], we pick 10 categories shared by *Office31* and *Caltech256* [10] as target domains. (3) *ImageNet-Caltech* [4] is a large-scale object classification dataset that consists *ImageNet-1K* (I) [35] and *Caltech256* (C), which contains 1,000 classes and 256 classes respectively as target domains. Task donated as S→T means that S is the source domain and T is the target domain. For task I→C, the training set of *ImageNet-1K* is used as source domain and a subset of 84 classes from *Caltech256* is used as target domain. While for task C→I the *Caltech256* is used as target domain and we choose the same 84 classes from the validation set of *ImageNet-1K* as the target domain.

**Baseline models**. We compare our results with non-transfer ResNet-50 [11], UDA method CDAN+E [26] and 9 state-of-the-art PDA methods including IWAN [50], SAN [2], PADA [3],

Table 1: Accuracy for Partial Domain Adaptation on *Office-Home*

| Method | Ar→Cl | Ar→Pr | Ar→Rw | Cl→Ar | Cl→Pr | Cl→Rw | Pr→Ar | Pr→Cl | Pr→Rw | Rw→Ar | Rw→Cl | Rw→Pr | Avg. |
|---|---|---|---|---|---|---|---|---|---|---|---|---|---|
| ResNet-50 [11] | 46.33 | 67.51 | 75.87 | 59.14 | 59.94 | 62.73 | 58.22 | 41.79 | 74.88 | 67.40 | 48.18 | 74.17 | 61.35 |
| CDAN+E [26] | 47.52 | 65.91 | 75.65 | 57.07 | 54.12 | 63.42 | 59.60 | 44.30 | 72.39 | 66.02 | 49.91 | 72.80 | 60.73 |
| IWAN [50] | 53.94 | 54.45 | 78.12 | 61.31 | 47.95 | 63.32 | 54.17 | 52.02 | 81.28 | 76.46 | 56.75 | 82.90 | 63.56 |
| SAN [2] | 44.42 | 68.68 | 74.60 | 67.49 | 64.99 | 77.80 | 59.78 | 44.72 | 80.07 | 72.18 | 50.21 | 78.66 | 65.30 |
| PADA [3] | 51.95 | 67.00 | 78.74 | 52.16 | 53.78 | 59.03 | 52.61 | 43.22 | 78.79 | 73.73 | 56.60 | 77.09 | 62.06 |
| MWPDA [14] | 55.39 | 77.53 | 81.27 | 57.08 | 61.03 | 62.33 | 68.74 | 56.42 | 86.67 | 76.70 | 57.67 | 80.06 | 68.41 |
| ETN [4] | 59.20 | 77.03 | 79.54 | 62.92 | 65.73 | 75.01 | 68.29 | 55.37 | 84.37 | 75.72 | 57.66 | 84.50 | 70.45 |
| DRCN [19] | 54.00 | 76.40 | 83.00 | 62.10 | 64.50 | 71.00 | 70.80 | 49.80 | 80.50 | 77.50 | 59.10 | 79.90 | 69.00 |
| AFN [47] | 58.93 | 76.25 | 81.42 | 70.43 | 72.97 | 77.78 | 72.36 | 55.34 | 80.40 | 75.81 | 60.42 | 79.90 | 71.83 |
| SLM [37] | 56.54 | **83.75** | **90.40** | **76.03** | 73.99 | 80.95 | 72.97 | 56.60 | 87.32 | **82.55** | 59.76 | 82.52 | 75.29 |
| BA³US [23] | 60.62 | 83.16 | 88.39 | 71.75 | 72.79 | 83.40 | 75.45 | 61.59 | 86.53 | 79.25 | 62.80 | 86.05 | 75.98 |
| Ours + BA³US | **64.66** | 82.97 | 89.12 | 75.67 | **75.52** | **85.36** | **78.51** | **64.24** | **88.07** | 81.27 | **65.31** | **86.67** | **78.20** |

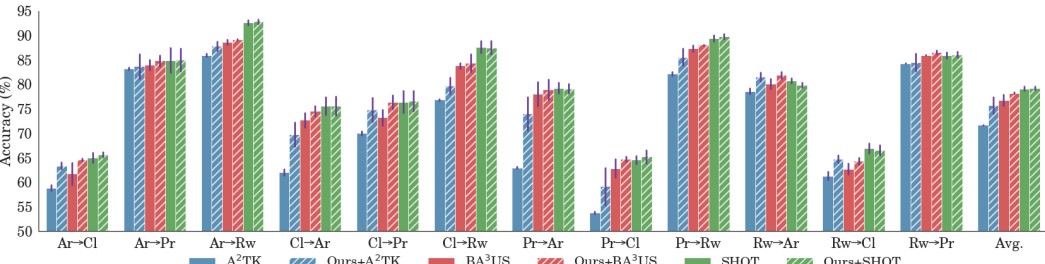

Figure 2: Performance of our implicit semantic response alignment on different backbones.

MWPDA [14], ETN [4], DRCN [19], AFN [47], SLM [37] and BA³US [23]. We use the classification accuracy for evaluating the performance. The classification accuracy is computed for every partial domain adaptation task and the average accuracy across all tasks is reported for each dataset.

**Implementation details**. We implement our model in PyTorch [32] using one NVIDIA Titan V GPU card. Our implementation on the purposed implicit semantic alignment module is added on the state-of-art PDA model BA³US [23]. For all three datasets, we adopt the pre-trained ResNet-50 [11] network as the backbone feature extractor. ASn auto-encoder with one hidden layer that consists of encoder $f$ and decoder $h$ is used as the class2vec machine. For each implicit semantic topic, we employ a multi-layer perceptron regressor with $l_1$ regulation as the attention receptor $g_j$. We train the network with the standard stochastic gradient descent optimizer and the learning rate is set to 1e-3 initially and decay exponentially during training. The learning rate of the backbone feature extractor is 0.1 of other layers and the topic attention receptors do not back-propagate to other parts of the model. $\alpha$ and $\beta$ are both set to 1 for *Office-Home* and *ImageNet-Caltech*, while for *Office31* we set $\alpha$ and $\beta$ to 0.1 and 0.5. $\lambda_{reg}$ is set to 0.5 in all experiments. For *Office-Home*, *Office31* and *ImageNet-Caltech*, the maximum iterations for training is set to 8,000, 4,000 and 40,000, respectively. The numbers of implicit semantic topics are set to 256, 64 and 16 separately for *Office-Home*, *Office31* and *Imagenet-Caltech*. We run our model five times with different random seeds and report the average classification accuracy in the following section. We also add our method on two additional PDA models, SHOT [22] and A²TK [17] to evaluate our method's generalization ability on different backbone PDA frameworks. Our code will be publicly available at: *https://github.com/implicit-seman-align/Implicit-Semantic-Response-Alignment*.

## 4.2 Algorithmic Performance

Here we comprehensively evaluate our proposed method with several state-of-art PDA models. Table 1 and 2 show the classification accuracy of 12 models on three benchmarks, where the best and second best results are bold and underline highlighted in the table. Some results are directly reported from BA³US [23] with the same protocol.

On the *Office-Home* dataset, ours method added with BA³US achieves the best or second-best results on 11 out of 12 transfer tasks. Regarding the average accuracy, our method advances the state-of-art results obtained by the original BA³US by 2.22% on this dataset. Compared with BA³US, our method makes more than 2% improvement on 9 tasks and the accuracy only decreases on one specific task Ar→Pr. Notice that for *Office-Home*, the state-of-art vanilla unsupervised domain adaptation

Table 2: Accuracy for Partial Domain Adaptation on *Office31* and *ImageNet-Caltech*

| Method | *Office31* | | | | | | | *ImageNet-Caltech* | | |
|---|---|---|---|---|---|---|---|---|---|---|
| | A→D | A→W | D→A | D→W | W→A | W→D | Avg. | I→C | C→I | Avg. |
| ResNet-50 [11] | 83.44 | 75.59 | 83.92 | 96.27 | 84.97 | 98.09 | 87.05 | 69.69 | 71.29 | 70.49 |
| CDAN+E [26] | 77.07 | 80.51 | 93.58 | 98.98 | 91.65 | 98.09 | 89.98 | 72.45 | 72.02 | 72.24 |
| IWAN [50] | 90.45 | 89.15 | 95.62 | 99.32 | 94.26 | 99.36 | 94.69 | 78.06 | 73.33 | 75.70 |
| SAN [2] | 94.27 | 93.90 | 94.15 | 99.32 | 88.73 | 99.36 | 94.96 | 77.75 | 75.26 | 76.51 |
| PADA [3] | 82.17 | 86.54 | 92.69 | 99.32 | 95.41 | 100.00 | 92.69 | 77.03 | 70.48 | 73.76 |
| MWPDA [14] | 95.12 | 96.61 | 95.02 | 100.00 | 95.51 | 100.00 | 97.04 | - | - | - |
| ETN [4] | 95.03 | 94.52 | 96.21 | 100.00 | 94.64 | 100.00 | 96.73 | 83.23 | 74.93 | 79.08 |
| DRCN [19] | 86.00 | 88.05 | 95.60 | 100.00 | 95.80 | 100.00 | 94.24 | 75.30 | 78.90 | 77.10 |
| SLM [37] | 98.73 | 99.77 | 96.1 | 100.00 | 95.89 | 99.79 | 98.38 | 82.31 | 81.41 | 81.86 |
| BA³US [23] | 99.36 | 98.98 | 94.82 | 100.00 | 94.99 | 98.73 | 97.80 | 84.00 | 83.35 | 83.68 |
| Ours + BA³US | 98.73 | 99.32 | 95.41 | 100.00 | 95.41 | 100.00 | 98.15 | 85.28 | 83.73 | 84.50 |

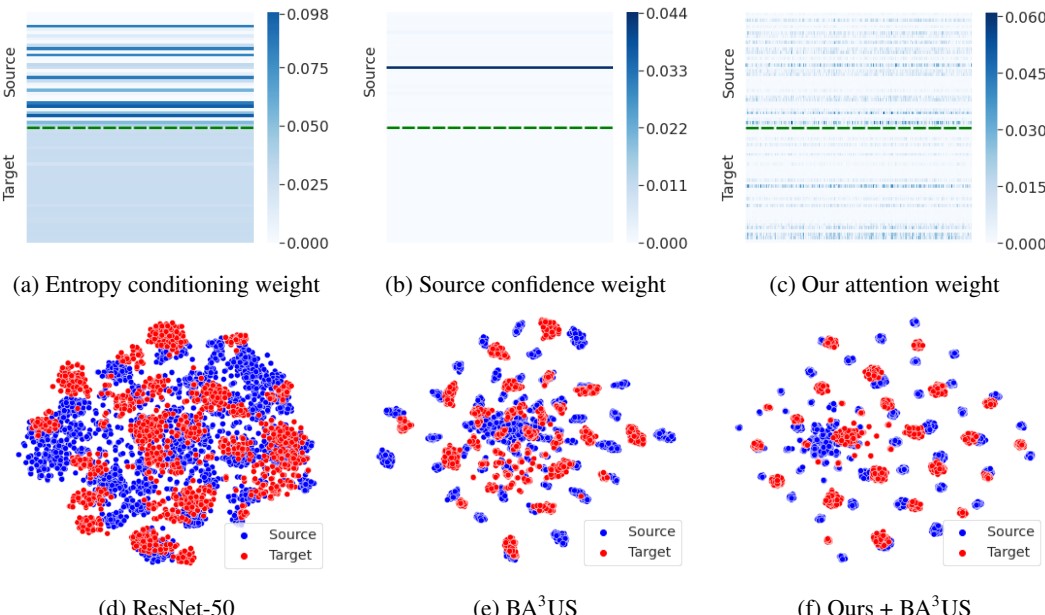

(a) Entropy conditioning weight    (b) Source confidence weight    (c) Our attention weight

(d) ResNet-50    (e) BA³US    (f) Ours + BA³US

Figure 3: (a-b) Entropy conditioning weight and source confidence weight of BA³US in one mini-batch of task the Ar→Cl ; (c) Ours attention map for the same mini-batch; (d-f) t-SNE visualization of features from Resnet-50, BA³US and ours of task Ar→Cl.

method CDAN+E performs even worse than ResNet-50, which demonstrates that the inconsistent label space, if not controlled with specific algorithmic design, inevitably brings in negative transfer by conventional domain adaptation algorithm. Some PDA methods including IWAN and PADA only slightly outperform ResNet-50. This implies the challenge posed by the negative transfer for this dataset and the improvement made by our method indicates the efficiency of implicit semantic alignment in this difficult setting.

Our method obtains the second-best average accuracy on the *Office31* dataset, but still slight improves the results of BA³US in 5 out of 6 tasks. *Office31* is a small-scale dataset with 31 categories, which makes it difficult to extract implicit semantic topics shared across different categories. Moreover, the high accuracy of BA³US also indicates that the shared categories already provide enough information for partial domain adaptation. On the other hand, our method achieves the best results for average accuracy and both transfer tasks on the large-scale *ImageNet-Caltech* dataset. Specifically, adding our model to BA³US boosts the accuracy by 1.28% for transfer task I→C, where the source domain contains a large number of irrelevant samples.

Our proposed implicit semantic response alignment is a plug-in module, which can be easily adapted to other domain adaptation backbones. Besides BA³US, we further evaluate our method with two additional state-of-art PDA models SHOT [22] and A²TK [17] on *Office-Home*. SHOT is a source-data-free domain adaptation framework that performs well on *Office-Home* in the partial setting. To

adapt SHOT in our scenario, we saved the 2048-dimensional source features after training on the source domain. During the training, we made the target model to learn semantics that are shared with the fixed source features. $A^2$TK is a PDA framework that directly trains on the fine-tuned ResNet-50 features. We run our experiment using the authors' code with fine-tuned features provided by this GitHub repository [45] and report our best results of $A^2$TK. As shown in Figure 2, adding our method on SHOT slightly improves the results. We conjecture that SHOT forces its target feature extractor focusing only on features that are similar to the source domain without adversarial domain alignment, which makes implicit semantic alignment less effective in the original space. Yet for $A^2$TK, our method significantly advances the average prediction accuracy by 3%.

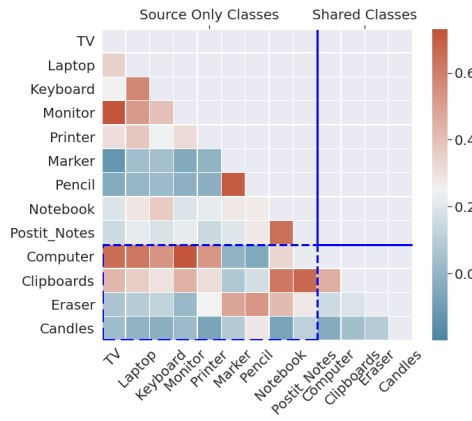

Figure 4: Category similarity

Table 3: Partial domain adaptation on individual class for task Ar→Cl on *Office-Home*

| Class | $n_s$ | $n_t$ | BA$^3$US | Ours | Improv.(%) |
|---|---|---|---|---|---|
| Computer | 99 | 44 | 12.12 | 59.60 | 47.48 |
| Clipboards | 40 | 25 | 67.50 | 87.50 | 20.00 |
| Eraser | 40 | 18 | 0.00 | 0.00 | 0.00 |
| Candles | 99 | 76 | 79.80 | 70.71 | -9.09 |
| All classes | 2427 | 1675 | 60.62 | 64.66 | 4.04 |

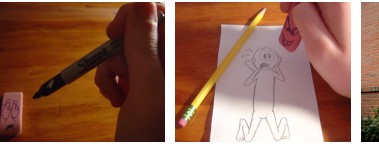

(a) Eraser      (b) Eraser      (c) Eraser

Figure 5: Example of confusing annotations

## 4.3 In-depth Exploration

In this part, we hereby discuss some properties of our purposed method. Specifically, we look into our weighting schema on the feature level and its effect on the feature space. Furthermore, we explore the relationship between classes to help understand how implicit semantic alignment impacts classification results in the scenario of partial domain adaptation. Finally, we elaborate the key hyperparameter analyses within our method.

**Topic attention weighting**. In Figure 3, we first showcase the feature-level attention weights with the weighting strategy of BA$^3$US [23] and ours on task Ar→Cl on *Office-Home*. Figure 3 (a) and (b) show the entropy-aware weights and source confidence weights of BA$^3$US on the features of one mini-batch during training. By assigning weights to samples and classes, the model suppresses the influence of negative transfer caused by irrelevant source samples. In Figure 3 (c), we visualize our attention weights corresponding to one implicit semantic topic on features of the same mini-batch. As expected, our weighting schema discoveries the information that responds to the same implicit topic in source and target samples on the feature level, which enables us to coordinate two domains based on the semantic topics, and transfers the effective information from multiple classes both the partial instance and feature levels. Moreover, we provide the t-SNE [43] visualizations of the final features of ResNet-50, BA$^3$US and ours in Figure 3 (d-e). The different distribution of feature embeddings demonstrates that BA$^3$US aligns the source and target samples with respect to classes, while our purposed method further divides the class-related clusters into smaller and well-separated clusters corresponding to the implicit semantic topics.

**Cross-class interaction**. We further explore the relationship between the target classes $\mathcal{C}_t$ and the source only classes $\mathcal{C}_s/\mathcal{C}_t$, as well as the impact of our purposed method on the cross-class interaction. For the task Ar→Cl on *Office-Home*, our method improves the accuracy in 11 out of 25 classes, where only 5 classes suffer from the performance drop. To check the relationship between classes, we draw the similarity matrix among the classes in the source domain with ResNet-50 features, then demonstrate 4 shared classes from $\mathcal{C}_t$ and 9 irrelevant classes from $\mathcal{C}_s/\mathcal{C}_t$ in Figure 4, where the relationship between shared classes and irrelevant source classes are highlighted with blue dashed lines in the bottom left corner. The per-class accuracy differences between our method and BA$^3$US for these 4 shared target classes are shown in Table 3.

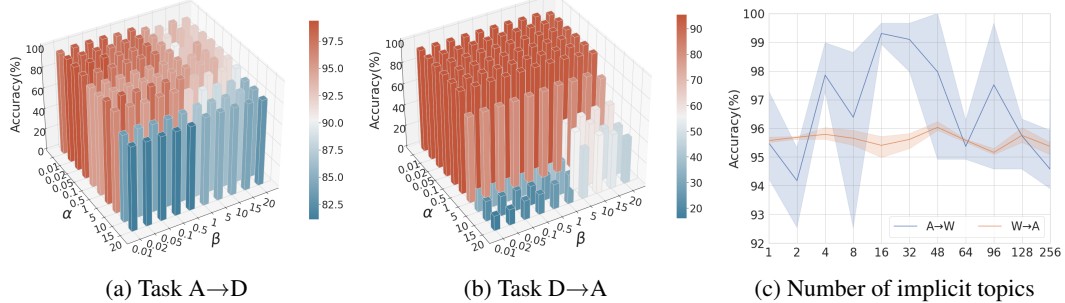

| (a) Task A→D | (b) Task D→A | (c) Number of implicit topics |

Figure 6: Parameter analysis on *Office31*, where (a-b) Performance with different $\alpha$ and $\beta$ on task A→D and D→A; (c) Performance with different implicit semantic topics on task A→W and W→A.

Our method significantly elevates the performance of categories Computer and Clipboards by 47.48% and 20.00%, respectively. The heatmap shows that these two classes are highly correlated to some of the extra source classes that are also semantically similar to Computer and Clipboards in real-world scenario. This huge performance boost illustrates that our method indeed extracts useful information for irrelevant source samples guide by implicit semantic alignment, promoting positive transfer in PDA setting. On the other hand, the prediction accuracy of Candles class decreases by 9.09%, because there is no valuable information in the unrelated samples for this class. Thus, forcing semantic alignment on this class could cause more severe negative transfer. The class Eraser is a failure case that we highlight here. It is very similar to Marker and Pencil classes in the feature space, but does not benefit from our method. We look into the original images and notice that the annotations for some Eraser samples could be quite confusing in this dataset. As shown in Figure 5, all three samples containing multiple objects are only labeled as Eraser; however, the main objects in these images should belong to Marker in Figure 5(a) or Pencil in Figure 5(b-c). In this case, our attention mechanism might be ineffective due to the confusing annotations in the dataset, since the feature extractor mainly focuses on the salient object instead of eraser itself.

**Hyperparameter analysis**. We explore our model's sensitivity to hyperparameters on four different tasks on *Office31*. Our loss function contains $\alpha$ and $\beta$ to balance the reconstruction loss and the alignment loss. As shown in Figure 6 (a) and (b), the performance decreases significantly when the ratio between $\alpha$ and $\beta$ grows. In this case, our model focuses only on extracting implicit topics, while ignoring the alignment. Another hyperparameter is the number of implicit topics $d_e$, which potentially controls the amount of semantic information extracted by our model. Figure 6 (c) stipulates that a small number works best for this uncomplicated dataset, because increasing the number of topics may force our model to align irrelevant noise. Notice that the results with different numbers of implicit topics in task A→W are inconsistent. After a in-depth exam of the dataset, we believe the inconsistency are caused by the nature of the source data. The Amazon domain in *Office31* has more heterogeneous images than the Webcam domain. In each category of the Amazon domain, images from the same category are of different objects. For example, the bike category in Amazon domain contains 82 images of 82 different bikes, while Webcam domain only contains 21 images of 6 unique bikes. Figure 6 (c) shows that the results of task W→A are consistent, when Webcam is used as source domain. As for more challenging datasets, experiment results indicates that a larger number improves the performance by including more semantic information. For $\lambda_r eg$, As we discussed in section3.2, it's only used for controlling the sparsity of the gradients in topic attention receptors, which are trained independently from other components in our model. In practice, this hyper-parameter does not have a significant impact on the performance and we empirically set $\lambda_r eg$ to 0.5 in our experiments.

## 5 Conclusion

In this paper, we presented Implicit Semantic Response Alignment, a novel approach to boost the existing partial domain adaptation models by exploring inherent class relationship across both source and target domains. It uncovered implicit semantic topics from features extracted by backbone model, and exploited a weighting strategy on the feature level with the help of topic-specific attention responses. The semantic-based alignment on the weighted features retained the relevant cross-domain

information contained in multiple categories. Extensive experiments on several partial domain adaptation benchmarks evinced the effectiveness of our method over state-of-art methods.

**Methodological Limitations**. The basic assumption of our implicit semantic response alignment is that different categories have partial in common in the implicit semantic space. By transferring the knowledge in the semantic level, rather than the category level, the relevant information from multiple categories are utilized to boost the performance. However, if there is little in common among different categories, such as cat and computer, our method might not bring in the improvements.

**Broader Impact**. Partial domain adaptation is a practical setting in unsupervised domain adaptation, where the source label space subsumes the target label space. In this paper, we explore the relationship among different categories and present it by implicit semantics. Based on this, we propose the implicit semantic response alignment to boost the positive transfer. The methodological philosophy on category relationship exploration can be adapted to tackle to other research problems, including information retrieval, recommendation systems, user modeling and so on.

**Potential Negative Societal Impact**. We address the well-defined partial domain adaptation problem and conduct experiments on benchmark dataset. It does not involve sensitive attributes and we do not notice any societal issues.

**Funding** This paper is partially supported by NSF OAC 1920147.

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
