# Implicit Semantic Response Alignment for Partial Domain Adaptation (Supplementary Material)

**Wenxiao Xiao**
Department of Computer Science
Brandeis University
Waltham, MA 02451
`wenxiaoxiao@brandeis.edu`

**Zhengming Ding**
Department of Computer Science
Tulane University
New Orleans, LA 70118
`zding1@tulane.edu`

**Hongfu Liu**
Department of Computer Science
Brandeis University
Waltham, MA 02451
`hongfuliu@brandeis.edu`

This supplementary material provides the experiment results for *DomainNet* [3] and *Visda-2017* [4] in partial domain adaptation (PDA), in responding to the reviewers' suggestion.

## Algorithmic Performance on *DomainNet* and *Visda-2017*

Here we reported the performance of our method on *DomainNet* and *Visda-2017* compared with BA[3]US [1], to demonstrate our performance improvements on large-scale datasets. *DomainNet* is a dataset of common objects in six different domain. All domains include a great number (345) of categories of objects such as Bracelet, plane, bird and cello. We only take three domains for our experiments: "real" (Rl), "painting" (Pt) and "clipart" (Cl), since we think these three domains are most likely to share similar semantics. To the best of our knowledge, there is no available source/target split for partial domain adaptation setting on DomainNet. We followed the data protocol in CuMix [2]. All 345 classes are used as the source domain and we select 100 classes with at least 40 images per category and non-overlapping with ImageNet as our target domain. The category list for the target domain is available at the paper's *github repository*. The large-scale *Visda-2017* dataset is a synthetic-to-real dataset for domain adaptation with over 280,000 images across 12 categories in the training, validation and testing domains. We take the "synthetic" (S) training domain and the "real" (R) validation domain, and select the first 6 categories (in alphabetic order) within each domain as partial target domain. We adopt the same hyperparameters as *Office-Home* in following experiments.

Table 1: Accuracy for Partial Domain Adaptation on *DomainNet* and *VisDa-2017*

| Method | DomainNet | | | | | | | VisDa-2017 | | |
|---|---|---|---|---|---|---|---|---|---|---|
| | Cl→Pt | Cl→Rl | Pt→Cl | Pt→Rl | Rl→Cl | Rl→Pt | Avg. | S→R | R→S | Avg. |
| BA[3]US [1] | 30.35 | 49.86 | **43.53** | 54.02 | 57.5 | 60.98 | 49.24 | 67.96 | 61.41 | 64.69 |
| Ours + BA[3]US | **34.33** | **51.24** | 42.14 | **54.04** | **61.99** | **61.22** | **50.83** | **70.67** | **66.14** | **68.41** |

The results of *DomainNet* and *Visda-2017* are shown in Table1, where results benefit from our method are bold highlighted in the table. Our method outperforms BA[3]US in 5 out of the 6 tasks on the challenging *DomainNet* dataset and increases the average accuracy by 1.6%. Our method has better accuracy in both tasks and achieves 4% average performance gain on *Visda-2017*. We also checked the per-class accuracy difference for task S→R. As expected, class car, which is semantically similar to three extra source classes (motorcycle, train and truck), gets a 34.59% boost (from 52.79% to

87.38%). Whereas class horse suffers a 17.01% loss (from 83.18% to 66.17%), since the horse category is the only animal class in this dataset.