# OpenReview forum: "Implicit Semantic Response Alignment for Partial Domain Adaptation"
_NeurIPS.cc/2021/Conference — NeurIPS 2021 Poster_

### Official Review · Reviewer_BdsE · 2021-07-13

**Rating:** 5
**Confidence:** 5

**Summary:**

This paper considers the problem of partial domain adaptation. To address this problem, the authors propose the Implicit Semantic Response Alignment (ISRA) method, which attempts to align the source and target domains on the semantic topic level instead of class level. Experiments on several datasets show the benefit of the proposed ISRA.

**Limitations And Societal Impact:**

Yes.

**Main Review:**

Pros:

+ This paper is well-written and easy to follow. The motivation and the overall organization are clear.

+ The proposed method is novel. As far as I know, this is the first work that tries to align the source and target domains in the semantic topic level. Also, the proposed method achieves good results on several datasets, and, it can be injected to different methods, such as SHOT, BA3US.

+ The authors provide good experiments to verify the effectiveness of the proposed method and also provide some analyses to reflect why the proposed method can work well.

Cons:

- Although the proposed method considers the partial domain adaptation, other settings, such as closed-set and open-set, also share the same semantic topics between the source and target domains. Therefore, I think the proposed method can also be applied to other settings. I would like to see whether the proposed method can improve the results on other settings. If not, please explain the probable reason.

- In hyperparameter analysis, the authors use office-31 to validate the parameters. However, since (1) the improvement on the office-home is not consistent and (2) the number of classes is limited, the results on office-31 can not well-reflect the parameter sensitivity of the proposed. Please provide the experiments on the office-home and the avg results on all directions should be reported.

- In addition, the proposed method seems sensitive to the number of implicit topics. We can find that, even using a small value, the results are not consistent for A->W setting.

- SHOT is a source-free method, how can the proposed method learn the semantic embedding for both the source and target domains?

- Why not evaluate the proposed method on the VisDa and DomainNet? DomainNet has a lot of classes and the classes will share more semantic topics, which is more suitable for the proposed method. Because the authors argue that the poor results on the office-31 are caused by the limited number of classes. Thus, when there are more classes (such as DomainNet), the proposed method will introduce more improvements.

- There are a lot of duplicate words between the Introduction and the motivation of the Method. Please modify one of them.

- In Fig~1, it would be better to show the details of the implicit semantic response alignment, including the structure of Auto-Encoder, the alignment process. Also, for better understand the proposed method, the author can provide an algorithm part.

Overall:

In summary, this paper proposes a novel method for partial domain adaptation and achieves promising results. However, some important experiments and explanations are missed in the current version. Therefore, I will give borderline reject this time and would like to see the response of the authors.


**Time Spent Reviewing:**

8 hours

---

> ### Author Response · Authors · 2021-08-09
> **Response to Reviewer BdsE**
>
> We are really appreciated the reviewer's recognition of the novelty and efficacy of our methods. In the following, we provide point-to-point responses. Especially, abundant extra experiments are added according to the reviewer's suggestions.
>
> **Closed-set and open-set** Great point! Actually the reviewer and we are on the same page. The key idea of our method is to transfer the same semantic topics between the source and target domains. Therefore, our proposed method can also be applied to other settings including the close-set and open-set settings. We did have some preliminary results on these before the submission and would like to share the results and analyses here.
>
> For the close-set setting, the backbone model BA$^3$US provides its results on *Office-Home*, where the authors of BA$^3$US replaced its adversarial loss with the CDAN's adversarial loss. We added our components to their BA$^3$US+CDAN model and reported our results in Table 1. Among 12 tasks, 6 of them benefit from our method and the average accuracy is improved by 0.5%. In close-set domain adaptation, the backbone model already utilizes enough samples in both domains. Our method could help further align the feature spaces of source and target domains on the semantic level, but resulting in a marginal performance gain.
>
> Table 1. Accuracy for close-set domain adaptation on *Office-Home*
>
> | | | | | | | | | | | | | | |
> |:---|:---:|:---:|:---:|:---:|:---:|:---:|:---:|:---:|:---:|:---:|:---:|:---:|:---:|
> |Method|Ar&rarr;Cl|Ar&rarr;Pr|Ar&rarr;Rw|Cl&rarr;Ar|Cl&rarr;Pr|Cl&rarr;Rw|Pr&rarr;Ar|Pr&rarr;Cl|Pr&rarr;Rw|Rw&rarr;Ar|Rw&rarr;Cl|Rw&rarr;Pr|Avg.|
> |BA$^3$US+CDAN|54.1|74.2|77.7|62.9|73.6|74.6|63.4|54.9|80.4|73.1|58.2|83.6|69.2|
> |Ours+BA$^3$US+CDAN|**54.6**|74.2|77.2|**63.1**|73.4|73.1|**65.9**|**57.3**|79.7|**74.9**|**59.3**|83.2|**69.7**|
> | | | | | | | | | | | | | | |
>
> For the open-set setting, we use SHOT as our backbone here since this model can handle the open-set settings, while the other two backbone models (BA$^3$US and A$^2$TK) cannot. Here we followed the setting of *Universal Domain Adaptation* (Reference [49]), where the shared categories are only a subset of both source and target label space. We reported the experimental results on *Office-Home* in Table 2. Our method improves the known category; but in the unknown and overall categories, our model performance is worse than the backbone. We conjecture the unknown target category contains multiple classes, which consist of shared but mixed-up semantics. Thus, this confuses the model to decide if a sample is from shared classes or the unknown class.
>
> Table 2. Accuracy for open-set domain adaptation on *Office-Home*
>
> | | | | | | | | | | | | | | |
> |:---|:---:|:---:|:---:|:---:|:---:|:---:|:---:|:---:|:---:|:---:|:---:|:---:|:---:|
> |**Known**|Ar&rarr;Cl|Ar&rarr;Pr|Ar&rarr;Rw|Cl&rarr;Ar|Cl&rarr;Pr|Cl&rarr;Rw|Pr&rarr;Ar|Pr&rarr;Cl|Pr&rarr;Rw|Rw&rarr;Ar|Rw&rarr;Cl|Rw&rarr;Pr|Avg.|
> |SHOT |66.05 |90.52 |97.68 |82.99 |79.98 |96.13 |84.29 |66.38 |97.22	|82.78 |71.51 |92.39 |83.99|
> |Ours+SHOT|**68.69**|**90.82**|**97.89**|**83.12**|**80.63**|95.98|**84.93**|**67.52**|**97.49**|**84.01**|**73.20**|**93.43**|**84.81**|
> | | | | | | | | | | | | | | |
> | | | | | | | | | | | | | | |
> |**Unknown**|Ar&rarr;Cl|Ar&rarr;Pr|Ar&rarr;Rw|Cl&rarr;Ar|Cl&rarr;Pr|Cl&rarr;Rw|Pr&rarr;Ar|Pr&rarr;Cl|Pr&rarr;Rw|Rw&rarr;Ar|Rw&rarr;Cl|Rw&rarr;Pr|Avg.|
> |SHOT |29.98 |32.24 |37.92 |51.78 |26.65 |33.67 |56.47 |37.92 |41.53 |57.48 |33.64 |34.00 |39.44|
> |Ours+SHOT |12.40 |11.81 |17.69 |43.94 |13.95 |16.01 |43.41 |13.51 |22.29 |44.30 |14.25 |12.64 |22.18|
> | | | | | | | | | | | | | | |
> | | | | | | | | | | | | | | |
> |**Overall**|Ar&rarr;Cl|Ar&rarr;Pr|Ar&rarr;Rw|Cl&rarr;Ar|Cl&rarr;Pr|Cl&rarr;Rw|Pr&rarr;Ar|Pr&rarr;Cl|Pr&rarr;Rw|Rw&rarr;Ar|Rw&rarr;Cl|Rw&rarr;Pr|Avg.|
> |SHOT |62.77 |85.23 |92.25 |80.15 |75.13 |90.45 |81.77 |63.80 |92.16 |80.48 |68.07 |87.08 |79.95|
> |Ours+SHOT |**63.58** |83.60 |90.50 |79.56 |74.56 |88.71 |81.16 |62.61 |90.63 |80.40 |67.84 |86.09 |79.10|
> | | | | | | | | | | | | | | |
>
> Based on the above results and analyses, we would like to focus on the partial domain adaptation setting, which mostly reflects the power of our implicit semantic alignment.
>
> **Hyperparameter** We would like to follow the reviewer's suggestion and add the hyperparameter analysis with task Cl→Rw on *Office-Home* dataset. Since we can only report tables in this response rather than figures, here we fixed one hyperparameter to 1, same as our *Implement details* section, and different values for another. In the revised version, we would like to add the hyperparameter analysis with task Cl→Rw in all directions in the figure visualization. The results are listed in Table 3. For this task, the best performance is achieved with $\\alpha$=0.1 and $\\beta$=1. The prediction accuracy is relatively stable for small $\\alpha$ and $\\beta$, but rapidly drops when the hyperparameters increases. We also tried the effect of topic number in task Cl→Rw. As shown in Table 4, the best number of the topics is 256 for this complicated dataset.
>
> Table 3. Parameters analysis on $\\alpha$ and $\\beta$ with task Cl→Rw on *Office-Home*
>
> | | | | | | | | | | | |
> |:---|:---:|:---:|:---:|:---:|:---:|:---:|:---:|:---:|:---:|:---:|
> |$\alpha$|0.01|0.02|0.05|0.1|0.5|1|5|10|15|20|
> |Accuracy|86.36|85.7|85.15|**87.08**|84.15|85.36|78.46|75.65|71.62|70.18|
> | | | | | | | | | | | |
> |$\beta$|0.01|0.02|0.05|0.1|0.5|1|5|10|15|20|
> |Accuracy|85.64|83.66|85.48|83.99|84.21|**85.36**|81.23|76.81|73.50|71.73|
> | | | | | | | | | | | |
>
> Table 4. Parameters analysis for the number of implicit topics $d_e$ with task Cl→Rw on *Office-Home*
>
> | | | | | | | | |
> |:---|:---:|:---:|:---:|:---:|:---:|:---:|:---:|
> |$d_e$|16|32|48|64|128|256|512|
> |Accuracy|83.89|83.39|83.45|84.72|84.99|**85.36**|83.89|
> | | | | | | | | |
>
> **Number of implicit topics** We conjecture that the inconsistent results with different numbers of implicit topics in task A→W are caused by the source data. The Amazon domain in *Offic31* has more heterogeneous images than the Webcam domain. In each category of the Amazon domain, images from the same category are of different objects. For example, the bike category in Amazon domain contains 82 images of 82 different bikes, while Webcam domain only contains 21 images of 6 unique bikes. Figure 6(c) shows that the results of task W→A are consistent, when Webcam is used as source domain. Thank you for pointing this out. We will add more analyses in the manuscript and make this point clear.
>
> **SHOT** Good catch! SHOT is a source-free domain adaptation method, which has excellent performances in several settings. We would like to see the performance of our method on SHOT. To adapt SHOT in our scenario, we saved the 2048-dimensional source features after training on the source domain. During the training, we made the target model to learn semantics that are shared with the fixed source features. We will make this point clear in the *Implementation details* section.
>
> **Performance on *VisDa-2017* and *DomainNet*** We agree with the reviewer that more classes might introduce more improvements when the extra classes are semantically similar to target classes. Here we reported the performance of our method on *VisDa-2017* and *DomainNet*. To the best of our knowledge, there is no available source/target split for partial domain adaptation setting on *DomainNet*. We followed the data protocol *Towards recognizing unseen categories in unseen domains* by Mancini, Massimiliano, et al. The category list for the target domain is available at *https://github.com/mancinimassimiliano/CuMix/tree/master/data/DomainNet*. All 345 classes are used as the source domain and we select 100 classes with at least 40 images per category and non-overlapping with ImageNet as our target domain.  Additionally, we only took three domains for our experiments: "real," "painting" and "clipart," since we think these three domains are most likely to share similar semantics. Note that due to the large size of these datasets, limited response period, and computational resources, both BA$^3$US and our method are not fine-tuned.
>
> Table 5. Accuracy for Partial Domain Adaptation on *DomainNet* and *VisDa-2017*
>
> | | | | | | | | | | | | | |
> |:---|:---:|:---:|:---:|:---:|:---:|:---:|:---:|:---:|:---:|:---:|:---:|:---:|
> | | | | |*DomainNet* | | | | \| | | *VisDa-2017* | |
> |Method| Cl&rarr;Pt |Cl&rarr;Rl|Pt&rarr;Cl|Pt&rarr;Rl|Rl&rarr;Cl|Rl&rarr;Pt|Avg.|\||Pr&rarr;Rw|Rw&rarr;Ar|Avg.|
> |BA$^3$US|30.35|49.86|43.53|54.02|57.5|60.98|49.24|\||67.96|61.41|64.69|
> |Ours+BA$^3$US|**34.33** |**51.24** |42.14  |**54.04**  |**61.99** |**61.22** |**50.83**| \| |**70.67**|**66.14**|**68.41**|
> | | | | | | | | | | | | | |
>
> The results of *DomainNet* and *VisDa-2017* are shown in Table 5. Our method outperforms BA$^3$US in 5 out of the 6 tasks on the challenging *DomainNet* dataset and increases the average accuracy by 1.6%. Our method has better accuracy in both tasks and achieves 4% average performance gain on *VisDa-2017*. We also checked the per-class accuracy difference for task S→R. As expected, class car, which is semantically similar to three extra source classes (motorcycle, train and truck), gets a 34.59% boost (from 52.79% to 87.38%). Whereas class horse suffers a 17.01% loss (from 83.18% to 66.17%), since the horse category is the only animal class in this dataset.
>
> **Other comments** Lastly, we thank the reviewer for the advice on more effective presentation of our method. We will remove the redundant parts in the motivation and add more detailed illustrations of our method, such as a clearer framework or an algorithmic flow.

---

### Official Review · Reviewer_SLrJ · 2021-07-16

**Rating:** 6
**Confidence:** 4

**Summary:**

This paper introduces a new partial domain adaptation (PDA) method. Different from previous methods that ignore the irrelevant source classes, this work argues that source data coming from extra classes also contributes to the adaptation. Motivated by this, the proposed method first identifies implicit semantic, then performs semantic alignment. The strength of this work is proved on common benchmarks.

**Limitations And Societal Impact:**

(1)	The motivation is not solid. Although the source examples from extra classes may share similar semantic information with target examples, these source examples also introduce noises and uncertainty.

(2)	Is that necessary to introduce another network (i.e., topic attention receptor) to get the attention by using its gradients? What is the motivation of this design? Why do we need this attention?

(3)	The section “Semantic topic alignment” is confusing. For example, which normalization method is used here? What is the formal definition of A?

Minor comments: In Line 155, it is more common to use Ct \subset Cs


**Main Review:**

Originality: Although the method is novel, its motivation still needs more clarification. The difference with previous contributions is clearly claimed.

Quality: This is a complete work but the proposed method is not technically sound (see details in Limitation).

Clarity: The writing of this paper needs further improvement. Furthermore, some points in the method section are not clearly explained (see Limitation).

I raise my rating to 6 based on the author's response and other reviewer's comments.

**Time Spent Reviewing:**

4 hours

---

> ### Author Response · Authors · 2021-08-09
> **Response to Reviewer SLrJ**
>
> Thank you for dedicating your time and energy into reviewing our paper. We would like to provide point-to-point responses as follows.
>
> **Motivation** Please allow us to be long-winded. We would like to re-introduce our motivation and then discuss the source noise issue.
>
> Our assumption of this paper is that different categories might share similar implicit semantics. Beyond the category level transfer, these implicit semantics can be utilized to boost the transfer performance by compensating extra knowledge in category level. For partial domain adaptation, most existing methods alleviate negative transfer by discarding/down-weighting the samples from irrelevant source categories. However, there are still valuable information in these extra categories, especially when relevant source classes contain limited samples. For example, extra source class dogs share semantic topics like fur and four legs with target class cats, while the dog class still has clear distinguished features like long noses. When classifying cat samples, the shared semantics can help separate cat from other target categories, such as birds or fish. To this end, our method attempts to utilize these meaningful semantics extracted from different parts of the extra sample images, instead of the whole images. On the other hand, irrelevant source semantics, like dog's long nose and noisy background, which are not shared by any target categories, will be suppressed with help of semantic topic alignment. Thus, this alignment process ensures that most noises and uncertainty will be alleviated.
>
> Admittedly, we agree with the reviewer that the source noises might still bring the negative transfer. We have discussed this issue in *In-depth Exploration* section. The below table (Table 3 in the manuscript) shows per-class accuracy differences between our method and BA$^3$US for these 4 shared target classes. Our method significantly elevates the performance of categories *Computer* and *Clipboards* by 47.48% and 20.00%, respectively. The category similarity in Figure 4 in the manuscript shows that these two classes are highly correlated to some of the extra source classes that are also semantically similar to *Computer* and *Clipboards* in real-world scenario. This huge performance boost illustrates that our method indeed extracts useful information for irrelevant source samples guide by implicit semantic alignment, promoting positive transfer in PDA setting. On the other hand, the prediction accuracy of *Candles* class decreases by 9.09%, because there is no valuable information in the unrelated samples for this class. This category suffers from the source noises. Thus, forcing semantic alignment on this class could cause more severe negative transfer.
>
> Table 3. Partial domain adaptation on individual class for task Ar→Cl on *Office-Home*
>
> | | | | | | |
> |:---|:---:|:---:|:---:|:---:|:---:|
> | Class | $n_s$ | $n_t$ | BA$^3$US | Ours | Improv.(%) |
> | | | | | | |
> | Computer | 99 | 44 | 12.12 | 59.60 | 47.48 |
> | Clipboards | 40 | 25 | 67.50 | 87.50 | 20.00|
> | Eraser | 40 | 18 | 0.00 | 0.00 | 0.00|
> | Candles | 99 | 76 | 79.80 | 70.71 | -9.09|
> | | | | | | |
> |All classes | 2427 | 1675 | 60.62 | 64.66 | 4.04|
> | | | | | | |
>
> **Topic attention receptor** Based on the above motivation, we proposed a novel framework that consists of implicit semantic discovery and semantic alignment. Since we expect to employ the semantics to boost the transfer performance, the first step is to semantic discovery, for which the topic attention receptor is designed. Specially, the topic attention receptor is introduced to identify which part (features) of a sample image represents a certain semantic topic. We used its absolute gradients as the attention signal, to guide our model only focus on the partial image (features), such as the regions containing fur or legs in a cat sample. The semantic topics represented by these partial regions are then evaluated by the following semantic topic alignment, which helps match the shared semantics and reduce discrepancy between the source and target domains.
>
> **Semantic topic alignment** The *Semantic topic alignment* section basically illustrates our approach of encouraging the same semantics from source and target domains as similar as possible. $A^{(j)} \\in \\mathbb{R}^{n \\times d}$ is formally introduced in Line 183, which has the same dimension as input feature $X$. It is the absolute gradients of the $i$-th topic attention receptor $g_i$ with respect to the input feature $X$. Then we applied $l_2$ normalization $N(A_j)$ to get the feature level weights, and finally got the masked feature corresponding to the $i$-th topic with element-wise multiplication $M_j = N(A_j) \\otimes X$, where $X$ denotes the source feature $X_s$ or target feature $X_t$.
>
> **Minor comment** We would like to use $C_t \\subset C_s$ in the revised version.
>
> **Summary** In the revised version, we will add the discussion on the noises and uncertainty in the motivation part, expand the explanation of framework overview and provide more details on semantic topic alignment.

---

### Official Review · Reviewer_BpRc · 2021-07-16

**Rating:** 8
**Confidence:** 4

**Summary:**

In this paper, the authors consider the partial unsupervised domain adaptation (PDA) problem. Different from the existing methods that assign lower weights to the source-specific classes and adapt one-to-one shared class mapping, the authors explore the relationship among different classes and expect to extract useful/similar information from other classes in a finer granularity for further boosting the positive transfer. The authors design a class2vec module to extract the implicit semantic topics from the visual features. Semantic responses of source and target data are aligned to retain the relevant information contained in multiple categories by weighting the features, instead of samples or classes. Extensive experiments on three benchmark datasets demonstrate the effectiveness of the proposed implicit semantic discovery and alignment.

**Limitations And Societal Impact:**

The limitations and societal impact are included in the main paper.
Please refer to the main review for suggestions in detail.

**Main Review:**

Strengths:
1. The idea of implicit semantic discovery and alignment seems interesting to me. The illustrative example in the Introduction helps to understand the common character among different classes, which can be utilized for further boosting knowledge transfer.
2. Inspired by the word2vec, the authors propose a similar method, class2vec, which presents each class into several implicit semantic topics. This module is an easy plug-in component, which can work with different PDA backbones to further improve their performance.
3. The technical details are clear and easy to follow. The authors propose the source code for easy use, which will benefit the community.
4. Experiments seem sufficient. The authors demonstrate the proposed technique on BA^3US has significant improvements compared with several SOTA baselines. The authors also demonstrate the effectiveness of implicit semantic discovery and alignment on different backbones.
5. The paper explains the results well. For example, Figure 3(c) provides how the proposed method works at the semantic level. It shows that the weighting schema discoveries the information that responds to the same implicit topic in source and target samples on the feature level, where only partial instances and partial visual features are activated to a certain implicit semantic topic. Figures 4&5 and Table 3 show when the implicit semantic discovery and alignment works or does not work. It helps when selecting a method for real-world applications.

Weaknesses:
1. For Figure 3(c), it is necessary to add the class category of each sample. By this means, we can tell whether some instances/categories share common semantics or not.
2. There are some presentation issues to improve the paper. For example, lambda_reg seems not discussed. The second best is not highlighted in Table 2 (W->D). Imagenet in line 229 should be ImageNet.

Concerns:
In Eq. (1), the authors extract the semantics at the instance level and aggerate them together to present the class level semantics. I am not sure whether adding another constraint that the instances from the same class share the same semantic would contribute to the positive transfer.


**Time Spent Reviewing:**

8

---

> ### Author Response · Authors · 2021-08-09
> **Response to Reviewer BpRc**
>
> Thank you for dedicating your time and efforts into reviewing our paper, and we sincerely appreciate your appreciation and recognition of the novelty of our work. Here we would like to address two weakness points and one open discussion from the reviewer.
>
> **Figure 3(c)** Good suggestion. Your insight on the visualization of our attention weights in Figure 3(c) is very helpful. We plan to annotate the samples that are most salient in this attention weight map and check if the categories of these annotated samples are indeed semantically similar.
>
> **Presentation** The presentation issues will be amended in our future version. As for $\\lambda_{reg}$, it is only used for controlling the sparsity of the gradients in topic attention receptors, which are trained independently from other components in our model. In practice, this hyper-parameter does not have a significant impact on the performance and we empirically set $\\lambda_{reg}$ to 0.5 in our experiments. The second best will be not highlighted in Table 2 (W->D). "Imagenet" in line 229 should be fixed to "ImageNet."
>
> **Open discussion** The suggestion of adding another constraint on the semantics to further promote the positive transfer makes sense to us. In the future, we would like to explore this direction by adding one constraint on the extracted semantic topics, and extend our work to other transfer learning settings.

---

### Official Review · Reviewer_9Z8X · 2021-07-17

**Rating:** 8
**Confidence:** 4

**Summary:**

This paper studies the partial domain adaptation problem. The authors propose an implicit semantic response alignment method where the source domain has more categories than the target domain. The motivation is to further utilize the extra source categories to boost the adaptation performance. To achieve this, anew module class2vec is proposed to extract the implicit semantics and explore the relationship among different categories, and align the semantic response of source and target data. Experiments on three benchmark datasets demonstrate the effectiveness of the proposed methods on different backbone networks.

**Limitations And Societal Impact:**

No major concerns on limitations or potential negative social impact.

**Main Review:**

Strong points
1.	The existing PDA methods design different strategies to lower the influence of source data that belongs to irrelevant classes. Despite the effectiveness of this practice, they inevitably overlook the potential” shared” knowledge among different categories, and treat the irrelevant class data as noise. Instead, the proposed implicit semantics idea investigated the “shared” knowledge across categories, which can be regarded as a concept of subclass, where different categories might share the same subclass.
I like this idea, since it is intuitive for partial domain adaptation. I think this idea might be also applied to other problems, like class-incremental continual learning, etc.
2. The presentation is clear and easy to follow.
3. The technical part is solid, although the semantic extraction is a simple auto-encoder. Considering this is the first paper to explore this direction, I am ok with it.
4. The proposed technique is an add-on module, which can be applied to the existing partial domain adaptation methods. The authors demonstrate the improvements with three partial domain adaptation methods, where the improvements on A^2TK and BA^3US are significant.
5. Experiments well validate the idea, including the standard algorithmic performance comparison and several in-depth explorations. The topic attention weighting and cross-class interaction are helpful to readers.

Weak points
1. Section 3.1 talks about motivation, which is duplicated with some content in the introductory part. It can be more concise.
2. Figure 4 is a good indicator to show why the proposed method works. It would be better if the authors can employ the category similarity to predict which category can be boosted. Some categories are dramatically enhanced while the candles category suffers from the negative transfer.

Typo: figure 1 caption, “purposed” -> “proposed”


**Time Spent Reviewing:**

4

---

> ### Author Response · Authors · 2021-08-09
> **Response to Reviewer 9Z8X**
>
> Thanks for your kind effort on reviewing our paper, and thanks for recognizing the novelty and the impact of our idea. Here we would like to address three weakness points from the reviewer.
>
> **Duplication content** We will modify the introduction and motivation sections to remove the redundancy.
>
> **Class-wise indicator** We appreciate your valuable advice on predicting performance gain based on category similarity. In the future, we plan to further explore the cross-class interaction and how to utilize this interaction to improve positive transfer as you suggested.
>
> **Typo** Thank you for pointing this out. We will modify it in the revised version.

---

### Decision · Program_Chairs · 2021-09-27

**Decision:**

Accept (Poster)

**Comment:**

This paper focuses on the partial domain adaptation problem. The proposal is an implicit semantic response alignment method where the source domain has more categories than the target domain. The philosophy behind sounds quite interesting to me, namely, utilize the extra source categories to boost the adaptation performance. This philosophy leads to a novel algorithm design I have never seen.

The clarity and novelty are clearly above the bar of NeurIPS. While the reviewers had some concerns on the significance, especially from Reviewer SLrJ and BdsE, the authors did a particularly good job in their rebuttal. Thus, most of us have agreed to accept this paper for publication! Please include the additional experimental results in the next version.